# Properties and Digestibility of Octenyl Succinic Anhydride-Modified Japonica-Type Waxy and Non-Waxy Rice Starches

**DOI:** 10.3390/molecules24040765

**Published:** 2019-02-20

**Authors:** Junhee No, Saehun Mun, Malshick Shin

**Affiliations:** 1Department of Food and Nutrition, Chonnam National University, Gwangju 61186, Korea; junheeno@naver.com; 2Center of Food and Bioconvergence, Seoul National University, Seoul 08826, Korea; saehun@snu.ac.kr

**Keywords:** japonica-type rice starch, octenyl succinic anhydride, amylose content, digestibility, emulsion stability

## Abstract

Waxy and non-waxy rice starches from japonica type Korean rice varieties were esterified with different levels of octenyl succinic anhydride (OSA), and the molecular structure of amylopectin (AP), digestibility, and emulsion stability were investigated. As OSA levels increased, the degree of substitution, granule size, peak and final viscosities, emulsion stability, and short chain of AP increased. However, the gelatinization temperature and enthalpy, and digestibility decreased. All OSA esterified starches showed a new band at 1723 cm^−1^, but maintained A-type crystallinity. The DP6-12 of AP in waxy rice starch (WRS) was higher than that in non-waxy rice starch (NRS) with increasing OSA levels. Because the amylose and long chain of AP accessed easily with OSA groups, the digestibility of NRS was lower than that of WRS. The emulsion stability was higher in WRS than in NRS. From the above results, it is suggested that amylose should have a higher affinity with OSA esterification than AP and that the emulsion stability should increase in WRS, but the digestibility should decrease in NRS after OSA modification.

## 1. Introduction

Starch consists of linear amylose (AM) and branched amylopectin (AP), and is the most important biopolymers due to its natural biodegradability, abundance, and annual renewal [1]. Most of native starches have been limited for use in food systems because of restriction by their properties. These include low shear strength, thermal instability, and high tendency of retrogradation. For these reasons, the modification of starch has developed to improve the limited aptitude of native starch through physical and chemical treatments. The modified starches are possible to use for tailored to specific food applications [2,3].

In 1953, the starch modification using octenyl succinic anhydride (OSA) first reported by Caldwell and Wurzburg has been applied to various fields for over half a century, especially as a food additive [4,5]. OSA esterified with starch through a standard process, in which octenyl succinic anhydride substituted with starch in mild alkaline solution. OSA starch has used as a good emulsifier owing to having hydrophobic groups, unlike native starch [2,6,7,8,9]. OSA starches combined with the hydrophobic and steric contributions of OSA can be not only used as a stabilizer and encapsulation agent, but also have different interfacial, thermal, nutritional, and rheological properties than native starches [5]. In the FDA, OSA starch approves as GRAS for food use and the upper limiting level of OSA is 3% per starch weight [6,7,10].

Rice, as a staple crop, is one of the main energy source in worldwide. Japonica and indica type rice are the two major domestic Asian rice varieties. Japonica type rice is cultivated in East Asia including Korea, whereas in most other regions indica rice is the dominant type of rice. Waxy and non-waxy rice differs the amylose content of starch. In general, waxy rice starch (WRS) has only amylopectin, but non-waxy rice starch (NRS) contains 7–33% amylose and 67–93% amylopectin [11,12]. Starch is the abundant ingredient of rice and starch granule size is the smallest and ranges from 2 to 8 μm. The tiny starch granules are caused to use in beverages and dressings, encapsulation of phytochemicals as a carrier in the delivery system, and as a fat replacer [13,14,15,16,17,18]. If rice starch has both hydrophilic and hydrophobic groups, it will also be available for emulsions and use in the pharmaceutical industry. The OSA-modified indica rice starches have been previously reported the physical and molecular properties [1,7,19,20]. A previous study on OSA-modified japonica rice starches was reported [14]. Japonica type Jinsumi rice starch was used as a lower amylose content compared to indica type Milyang 261 rice starch. Consequently, low amylose rice starch had higher stability and coating efficiency than high amylose starch. However, there is currently no information on characteristics of OSA-modified japonica-type rice starches.

Therefore, the objectives of this research were to prepare OSA-modified rice starches using japonica-type WRS and NRS, and to compare the differences between OSA modified WRS and NRS. The physicochemical properties, particle size distribution, crystalline structure, morphology, and molecular structures of amylopectin, in addition to digestibility and emulsion stability for application in food systems were compared two OSA rice starches.

## 2. Results and Discussion

### 2.1. DS of OSA-Modified Rice Starch

The DS of OSA rice starches prepared with different OSA levels were analyzed. The effect of the OSA levels was represented at the addition of 1, 2, and 3% OSA, while keeping constant all of the other conditions given above. The DS of WRS and NRS manufactured using 1, 2, and 3% OSA were 0.0140, 0.0246, and 0.0262, and 0.0147, 0.0213, and 0.0258, respectively. The DS of OSA rice starches increased with increasing levels of OSA. When a relatively lower level of OSA was added, the DS increased rapidly, whereas in higher levels, the DS increased slowly. These results were in agreement with other studies [7,21]. It was known that OSA molecules are esterified within the vicinity of starch hydroxyl groups, which are located in C-2, C-3, and C-6 of a glucose molecule under mild alkaline conditions [6,14]. For starches with a higher degree of amylopectin branching, such as waxy starches, Simsek et al. [17] reported that waxy starches had a lower DS due to steric factors. In case of indica rice, the amylose content showed a positive relation to OSA modification. The amylose molecules present in the amorphous domains of the starch granule was more accessible for esterification with OSA [20]. However, Jung & Youn [14] reported that the OSA-modified Jinsumi cultivar (15.42% amylose content) had a higher DS than the high amylose rice starch, Milyang 261 cultivar (20.31%). This result was consistent with our results. With the same level of OSA used, the OSA-modified WRS showed a higher DS than the OSA-modified NRS, except in the case of 1% addition of OSA. This indicates that other factors are affecting the DS, not just amylose content. No and Shin [18] reported that the DS of OSA-modified WRS, Hwaseonchal cultivar, increased with increasing levels of OSA, showing DS values of 0.0062, 0.0182, and 0.0214 corresponding to 1, 2, and 3% OSA levels, respectively. One of the other WRS, Sincheonchal starch, also showed a higher DS than Hwaseonchal starch, suggesting that it caused to difference between different waxy rice cultivars. This difference was due to the different molecular structures among starches. It was reported that the DS of OSA-modified starch was influenced by starch sources, cultivar, amylose content, concentration of sodium hydroxide, OSA levels, and mechanical activation time [8,22].

### 2.2. Particle Size Distribution and Morphology

#### 2.2.1. Particle Size Distribution

The particle size distribution (PSD) of OSA-modified WRS and NRS is shown in Figure 1. Both native and OSA-modified starches had a trimodal PSD, regardless of rice starches isolated from different cultivars. The mean particle size ranged from 3.28–6.13 µm. In the case of Hopyeong, when starch modified with OSA, the second peak decreased, the third peak increased, and the mean particle diameter gradually increased with the level of OSA added. Inserting an OSA group increased the mean particle diameter. Other studies have reported a size increase in granules subjected to OSA modification, partly attributable to the aggregation between starch granules caused by increased hydrophobicity [23].

However, in case of Sinseonchal WRS, it seemed that the addition of OSA had no noticeable effect on PSD, and the mean particle diameter started to increase when 2% OSA added. The OSA substitution with starch occurs mostly in the amorphous region of starch molecule without affecting the crystalline region, inserting an OSA group along amylose molecules and/or at branch points of AP [17,24]. Because OSA modification occurred in mild alkaline conditions (pH 8.5), the OSA group is preferentially bound to the AM and the branch point of AP in the amorphous region, rather than entering and binding to the crystalline region. These might be the reasons why there was a difference in mean particle diameter and PSD between OSA-modified WRS and NRS.

#### 2.2.2. Scanning Electron Microscopy (SEM)

The shape and size of OSA-modified rice starches was observed using SEM and compared to that of native starch (Figure 2). The OSA starch granule retained the granular morphology of native starch when the DS was 0.0140–0.0147, whereas when the DS was 0.0213, polygonal-shaped granules had tiny pores on the surface. When the DS increased to 0.0246 in WRS and 0.0258 in NRS, the surfaces of the OSA starch granules became rough, and the granules started to cluster. This clustering process might have due to the enlarged and disrupted starch granules occurring during esterification. The functional group of OSA accessed to the starch granular surface, and made the pores on the surface [7]. The granular sizes of Hopyeong NRS and Sincheonchal WRS were 3.28 µm and 4.07 µm, respectively. The particle size of the OSA-modified starch increased with increasing levels of OSA addition. According to previous studies, the size of amphiphilic nanoparticles increased because of the hydrophobic group [9,23]. The particle size of OSA-modified WRS was smaller than that of OSA-modified NRS. Generally, the stability of OSA starch is inversely proportional to particle size, because of smaller particles giving a higher packing efficiency. Therefore, this produces a more homogenous layer than larger particles, which should have higher detachment energies [25]. The OSA-modified Sinseonchal WRS showed smaller particles, and therefore the OSA-modified WRS may have potential for use as an encapsulation material.

### 2.3. Crystalline Structure

#### 2.3.1. Fourier Transform Infrared Spectroscopy

Octenyl succinylation is going to substitute the hydroxyl groups of starch molecules for carbonyl groups of OSA. The insertion of these carbonyl groups to starch molecule structure confirmed through FT-IR spectroscopy [7,8,16,17]. The FT-IR spectra of the native and OSA-modified starches are presented in Figure 3. The all rice starches have similar profiles, regardless of OSA modification. The broad peak appeared at approximately 3400 cm^−1^, indicating the presence of a hydroxyl group (O–H). The peaks at 2930 cm^−1^ and 1640 cm^−1^ observed the C–H stretching vibration and bound water present in the starch, respectively [17,26]. In the starch spectra, several unique peaks at 800–1200 cm^−1^ were due to stretching of the C–O bond. Compared to native starch, the spectrum of OSA-modified starch showed two new peaks at approximately 1572 and 1723 cm^−1^. The peak at 1723 cm^−1^ was attributable to the meaningful IR stretching vibration of C=O that occurred, indicating existing the ester carbonyl groups. Whereas the peak at 1572 cm^−1^ was revealed the asymmetric stretching vibration of carboxylate RCOO^−^ [4,7,21]. These results proven that native rice starches were modified successfully by OSA.

#### 2.3.2. X-ray Diffraction

The crystallinity of native and OSA-modified rice starches was measured using X-ray diffraction (XRD) pattern (Figure 3). Both rice starches had A-type crystallinity, showing peaks at diffraction angles (2θ) of 15.3, 17.1, 18.0, and 23.2°. It is suggested that esterification with OSA did not cause to the change in the crystalline pattern of starches up to DS 0.0262. As the esterification of starch with OSA mainly occurred in the amorphous region, the crystalline pattern of starch maintained after chemical modification [7,8]. However, the relative crystallinity of OSA-modified starch gradually decreased compared to the native starch, respectively (*p* < 0.05). It is possible that esterification of OSA occurred in amorphous regions of starch, might affect the crystalline region, resulting in the reduction of the crystalline region’s intensity [22].

### 2.4. Chain Length Distribution of Amylopectin

The percentages of chain length distribution of AP in native and OSA-modified rice starches are presented in Table 1. The average chain length of the OSA rice starches decreased slightly with OSA substitution. Hopyeong NRS was significantly different, with OSA levels in the DP ≥ 37 of amylopectin. Sinseonchal WRS had a significant increase in DP6-12 but a decrease in DP 25-36, and DP ≥ 37 after OSA substitution. The chain length of AP in Sinseonchal WRS affected by OSA modification than that in Hopyeong NRS. It could be explained the fact that OSA can be preferentially esterified with an AM molecule in Hopyeong NRS, due to its positive impact on reaction efficiency. As previously mentioned, OSA preferentially substituted within the amorphous region, which AM chains existed predominantly of starch granules [4]. The flexibility of AM chains allow to extend and form complexes with the OSA group over more than one AP molecule domain [27]. Consequently, the branched chain length of NRS was not affected by OSA modification compared to that of WRS.

Sinseonchal WRS was consisted only AP, the outer long-chains of AP can substitute for OSA during modification. Therefore, the chain length distribution of Sinseonchal AP changed more apparent than that of AP in Hopyeong NRS. When the distribution of OSA groups in OSA waxy maize starch was studied, the OSA groups of starch were distributed in the interior and amorphous parts of AP, as well as on the outside of the starch granules [17,24].

### 2.5. In Vitro Digestibility

The in vitro digestibility of native and OSA rice starches is shown in Table 1. OSA-modified starches showed that RDS value decreased with increasing OSA levels. Unlike the RDS values, the SDS values did not affect by OSA modification with different levels. The RS values of native NRS and WRS were 3.20% and 0.03%, respectively. OSA modification of rice starch significantly increased RS content but decreased RDS content compared to native starch, regardless of the botanical sources [17]. The digestibility of the OSA-modified rice starches decreased with increasing OSA levels, and the decreasing rate of NRS was higher than that of WRS. This difference probably has been due to the esterification between AM molecules of NRS. The linear AM molecules existed in amorphous region in starch granule and OSA esterified AM is difficult to hydrolyze with α-amylase. Hu et al. [8] reported the RS levels of OSA-modified indica rice starches increased with increasing AM contents within the same type of rice. This result was consistent with the results of our study.

### 2.6. Pasting and Thermal Properties

The pasting characteristics of OSA-modified rice starches are shown in Table 2. All pasting properties differed significantly according to the rice cultivars and OSA levels (*p* < 0.05). The pasting temperature of native NRS was higher than that of native WRS. The OSA-modified rice starches increased in peak, trough, and final viscosities, but decreased in pasting temperature with increasing OSA levels. The effect of hydrophobic groups in the OSA starches could increase the viscosity, according to the formation of a network of hydrophobic containing polymers such as increase hydrophobicity of cellulose by modification [28]. The paste of the OSA-modified starch was higher viscous than that of native starches [2,7,18,19,20,29]. The attachment of a bulky group such as OSA increased the pasting viscosities of the starches, so the modified starch had the extended tendency of paste viscosity. He et al. [20] suggested that OSA-modified starches reduced degrees of crystallinity slightly compared to native starches. Most pasting viscosities of OSA starches differed significantly by different cultivars. Only the setback viscosity of WRS was much lower than that of NRS. The WRS showed the higher peak viscosity than NRS. The increase of the OSA groups showed the increase of peak and breakdown viscosities. It was found that the AM molecules in starch granule could be come close with OSA groups easier than the AP molecules.

The onset, peak, and conclusion temperatures, as well as the gelatinization enthalpy of native and OSA-modified starches are shown in Table 2. The onset temperature tent to be lower than the pasting temperature by RVA. As the OSA modification was shifted the peak to lower temperatures and the size of peak to broader and smaller. The OSA groups of OSA starch weakened the orderly structure of the starch, in turn lowering the resistance to thermo-transition. As a result, the gelatinization temperature reduced with increased OSA substitution ratio [19].

The peak temperature of the OSA starches similarly affected the pasting temperatures from the RVA profiles. For instance, hydrophobic groups weakened the interactions between starch molecules, allowing the granules to gelatinize at lower temperatures [4]. These results were similar with those of previous reports, regardless of the botanical sources, consisting of: corn [6,10,30], tapioca [28], bananas [31], wheat [29], rice [7,14,20], or potatoes [30]. The gelatinization enthalpy decreased in OSA Hopyeong NRS but rapidly decreased in OSA Sinseonchal WRS. This result could explain that the hydrophobic alkenyl groups weakened the internal hydrogen bonding of starch, enabling the starch to gelatinize at relatively low temperatures, and hence gradually decreasing the enthalpy of OSA starches with increased DS [19,29]. The reduction of DP > 13 of AP (Table 1) led to the decreased T_o_, T_p_, and ΔH of the OSA-modified starches. The T_o_, T_p_, and ΔH of WRS all decreased more than those of NRS, due to the increment of the DP6-12 chain length proportion within the AP molecule to substitute with OSA groups.

### 2.7. Emulsion Stability and Oil Absorption Capacity

The OSA rice starches can use as an emulsifier due to the hydrophobicity of the octenyl group. The data of creaming index and oil absorption capacity of OSA-modified NRS and WRS were used as a measure of emulsion stability (Figure 4).

Creaming is defined the oil-in-water instability that originates from a density difference of two liquids in emulsion [32]. The creaming index have measured by the direct observation of emulsion separation using mass cylinder [32,33]. A lower creaming index value indicates higher emulsion stability. The emulsion containing Hopyeong native starch separated after 4 h, but the emulsion containing 3% OSA-modified starch did not separate until 16 h, indicating that modification of starch with OSA improved the emulsion stability. The creaming index decreased with the increasing DS of starches and storage times. Like NRS, OSA-modified WRS also decreased the creaming index with increased DS. However, an emulsion containing native WRS maintained a creaming index below 20%, and emulsion containing 3% OSA added WRS was not separated until stored for 48 h. This result might be caused to the higher viscosity of OSA-modified WRS compared to the OSA-modified NRS. Higher viscosity reduced the velocity of emulsion separation. This result suggested that OSA modification of starch gave good emulsifying properties to starches, regardless of cultivar and amylose content.

The oil absorption capacity of OSA-modified WRS and NRS ranged from 130.42–142.66% and 117.28–148.98%, respectively (*p* < 0.05). OSA-modified NRS showed better oil absorption capacity than OSA-modified WRS, because the AM molecules were more accessible to esterification with OSA. Amylose linear chain formed inclusion complexes with certain hydrophobic ligands, such as iodine, alcohols, fatty acids, and surfactants [4]. OSA modification increased the oil absorption capacity and hydrophobicity of the rice starches. The higher the oil absorption capacity was, the thicker the adsorbed layer was around the droplets [8].

OSA-modification of NRS and WRS purified from japonica-type Korean rice varieties showed different molecular size, DP of amylopectin and pasting properties, in vitro digestibility, and emulsion stability. The results should be due to the difference of AM content, but caused by differences in the site of OSA group attachment with AM, depending on the molecular structure and AM content. Generally, it is well known that AM molecule is more easily attached to the OSA group compared to AP molecule. However, the chain length distribution of AP also suggested that long chains of AP could be esterified with OSA groups.

## 3. Conclusions

Waxy and non-waxy rice starches from japonica-type Korean varieties were esterified with octenyl succinic anhydride (OSA) at different OSA levels, and the physicochemical, pasting, and thermal properties, the molecular structure of AP, digestibility, and emulsion stability were investigated. The DS increased with increasing OSA levels, and was not significantly different between WRS and NRS. The OSA-modified WRS showed an increase in DP6-12 of AP, peak and final viscosities, and emulsion stability, but showed a decrease in digestibility, DP ≥ 25 of AP, pasting temperature, and endothermic enthalpy. OSA modification of NRS increased peak, final, and setback viscosities, and granular size, but decreased digestibility, DP ≥ 37 of AP and pasting temperature. OSA-modified NRS showed higher granule size, and pasting temperature, but a lower digestibility, DP6-12 of AP than OSA-modified WRS. The emulsion stability increased with increasing DS in both starches. It may be that OSA undergoes earlier esterification with the AM molecule of NRS compared to AP of WRS. Additionally, as OSA-modified WRS showed smaller granule sizes compared with OSA-modified NRS, it could possibly be more useful to use wall materials for emulsifying and encapsulation.

## 4. Materials and Methods

### 4.1. Materials

Japonica type two Korean rice varieties, waxy Sinseonchal and non-waxy Hopyeong rice, were received from the National Institute of Crop Science, Rural Development Administration in Wanju, Jeollabukdo, Korea. Octenyl succinic anhydride (Milliken Chemical, Spartanburg, SC, USA) was obtained from Daesang Co., and canola oil (Beksul, CJ Cheil Jedang Corp., Seoul, Korea) was purchased. Pancreatin (P7545 from porcine) and amyloglucosidase (A9913), isoamylase and D-glucose GOPOD-format assay kit were purchased from Sigma-Aldrich Chemical Co. (St. Louis, MO, USA) and Megazyme (Wicklow, Ireland), respectively.

### 4.2. Purification of Rice Starch

Starch was isolated by the alkaline steeping method [34] from japonica type Korean rice varieties, waxy, Sinseonchal, and non-waxy, Hopyeong white rice grains. The grains were soaked and then ground with alkaline solution (0.2% NaOH) using a food grinder (Hanarossack, Daesung Altron Co., Seoul, Korea). After grinding, the slurry passed through a 100- and 270-mesh sieve continually, and the process repeated. The starch was collected by centrifuging at 1630 g for 10 min (Supra 22K, Hanil Science Industrial Co., Seoul, Korea), neutralized with 1 N HCl solution, washed, and centrifuged (2730 g for 10 min). The isolated starch was air-dried, ground (<150 μm), and stored until use. The amylose contents of Sinseonchal and Hopyeong starches were 0.24% and 16.69%, respectively [35].

### 4.3. Preparation of OSA-Modified Starch and Determination of DS

Rice starch (50 g, dry basis, db) and distilled water (140 mL) were added into three neck round bottom flask, mixed with magnetic stirrer to adjust pH 8.5 using 1 N NaOH. Octenyl succinic anhydride (1, 2, and 3% based on starch weight) added dropwise (0.1 mL/min) with continuous stirring (at pH 8.5). After modifying for 6 h, the starch reactant adjusted to pH 7, washed and centrifuged (2730 *g* for 10 min). Thereafter, the OSA starch dried at room temperature passed through a 100-mesh sieve. The degree of substitution (DS) was analyzed in accordance with the titrimetric method [4,6,13,29].

### 4.4. Particle Size Distribution

The particle size distribution of OSA starch was estimated using a Malvern Mastersizer 3000 (Malvern Instruments Ltd., Worcestershire, UK). The OSA rice starch was dispersed in distilled water (0.01%, *w*/*v*) with sonication at 500 W for 2 min.

### 4.5. Digestibility of OSA-Modified Rice Starch

In vitro digestibility of OSA starch was determined using the method outlined by Lv et al. [5]. Enzyme mixture which prepared with α-amylase from porcine pancreas and amyloglucosidase added into the starch (200 mg, db) in phosphate buffer (15 mL, 0.2 M, pH 5.8) with 10 glass beads (ϕ 4 mm) and the reactant was incubated in a 37 °C shaking water bath (160 rpm). Following 20 and 120 min of incubation, hydrolyzed fluid (0.5 mL) added absolute ethanol (4.5 mL) was centrifuged at 1630 g for 10 min. The liberated glucose was analyzed by the GOPOD method using a glucose assay kit. The percentages of rapidly digestible starch (RDS), slowly digestible starch (SDS), and resistant starch (RS) were calculated using the Englyst et al. method [36].

### 4.6. Morphology

The shape and size of OSA starch was observed using a scanning electron microscope (SEM, JAM-540, JEOL Tokyo, Japan). The OSA-modified starch sample put on a SEM stub and coated with gold/platinum under vacuum. The SEM acceleration conditions were voltage 15 kV for 100 sec, and the magnifications used were 2000× and 7000×.

### 4.7. Crystalline Structure Analysis

#### 4.7.1. Fourier-Transform Infrared Spectroscopy

The Fourier-transform infrared (FT-IR/NIR) spectra of the OSA rice starches were observed using an FT-IR/NIR spectrometer (Perkin Elmer, Waltham, MA, USA) at room temperature [37,38]. OSA starch was prepared as a KBr pellet and scanned against a blank KBr background within a frequency range of 4000–400 cm^−1^ under a 4 cm^−1^ resolution.

#### 4.7.2. X-ray Diffractometry

The crystallinity and intensity of the OSA starch were examined by an X-ray diffractometer (3D High Resolution X-ray Diffractometer, Empyrean, PANalytical Co., Almelo, Netherlands). The OSA starch was kept in a desiccator until the moisture removed. The diffractometer was operated under the following conditions; Cu-Kα; Ni; 3000 cps, 8°/min; diffraction angle (2θ) 5–40°; 30 mA; and 40 kV. The relative crystallinity of OSA starch was calculated following the method of Mun & Shin [39] using software (Origin 6.0, Microcal, Northampton, MA, USA).

### 4.8. Determination of Chain Length Distribution of Amylopectin

The branched chain length distribution of AP was analyzed by high-performance anion-exchange chromatography (HPAEC) with pulsed amperometric detection (HPAEC-PAD; ICS-5000, Dionex Co., Sunnyvale, CA, USA) [38,39]. Briefly, OSA starch (0.01 g, db) added in 90% DMSO (2 mL), solubilized with heating and precipitated with absolute ethanol. The starch solution, 2 mL of sodium acetate buffer (50 mM, pH 3.5) and isoamylase (5 μL) were mixed, and the mixture was reacted at 37 °C with stirring. After inactivating the enzyme with heating for 10 min, the 0.4 mL of solution solubilized into 240 mM NaOH (2 mL) and filtered by nylon syringe filter (0.45 μm). The filtered starch solution injected through the HPAEC-PAD system, consisting of an electrochemical detector and a CarboPac PA200 column (3 × 250 nm, Dionex Co.). A gradient eluent of 240 mM NaOH and 500 mM sodium acetate in 240 mM NaOH was used as a mobile phase at 0.5 mL/min.

### 4.9. Analysis of Pasting and Thermal Properties

The pasting behaviors of the OSA starch were investigated using a Rapid Visco Analyzer (RVA-TecMaster, Perten Instruments AB, Hägersten, Sweden). Starch (2.5 g, 12% mb) was mixed with distilled water (25 mL) and stirred using a constant paddle at 160 rpm/min. The sample canister held at 50 °C for 1 min, heated to 95 °C for 3.7 min, kept at 95 °C for 2.5 min, cooled down to 50 °C for 3.8 min, and maintained at 50 °C for 2 min [34]. The pasting temperature, peak (P), trough (T), final (F), breakdown (P-T), and total setback (F-T) viscosities were measured.

The thermograms of the OSA starches were obtained using a differential scanning calorimeter (DSC-Q1000, Universal V.3.6C TA Instruments, Olivia Gibson, UK) with calibration using indium. Starch (3.0 mg, db) and deionized water (6.0 mg) were put into the aluminum pan, sealed, allowed to equilibrate for 12 h, and programmed at a rate of 10 °C/min from 30 to 130 °C [40]. The onset (T_o_), peak (T_p_), and conclusion (T_c_) temperatures, and gelatinization enthalpy (∆H) were compared.

### 4.10. Determination of Oil Absorption Capacity and Emulsion Stability

The oil absorption capacity of the OSA starch was measured according to the method outlined by Bhosale and Singhal [6]. The OSA starch (0.5 g) mixed with oil (15 mL) for 30 sec using a vortex mixer. The mixture was left for 30 min at room temperature and was centrifuged at 2730 g for 20 min. The supernatant oil removed until no further oil separated by the difference in the weight of the sample following centrifugation. The oil absorption capacity was calculated by following equation:Oil absorption capacity (%) = (Sediment weight (g) − Sample weight (g) × 100/Sample weight (g)(1)

The emulsion stability of OSA starch was measured by the method of Krstonošić, Dokić, Nikolić and Milanović [32]. Five milliliters of refined oil and 5 mL distilled water with 0.04% xanthan gum were homogenized using a homogenizer (HG-15D, WiseTis, homogenizer, Daihan Scientific Co. Ltd. Wonju, Korea) at 15,000 rpm for 2.5 min to make an emulsion with 10% OSA starch. After the emulsion was stored at room temperature for 2 days, its stability (%) was calculated with the percentage of the separated layer of emulsion divided by total volume of emulsion.

### 4.11. Statistical Analysis

The results were represented as the mean ± standard deviation at least triplicate. The data were analyzed through the analysis of variance (ANOVA). SPSS 12.0 (SPSS Inc., Chicago, IL, USA) was used using a Duncan’s multiple range test for significance at *p* < 0.05.

## Figures and Tables

**Figure 1 molecules-24-00765-f001:**
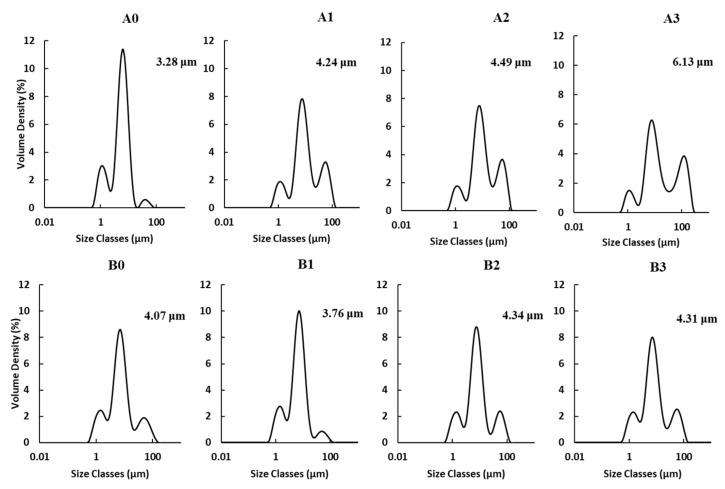
Particle size distribution of OSA-modified Hopyeong (A) and Sinseonchal (B) rice starches. 1, 2, and 3 refer to the added percentages of octenyl succinic anhydride.

**Figure 2 molecules-24-00765-f002:**
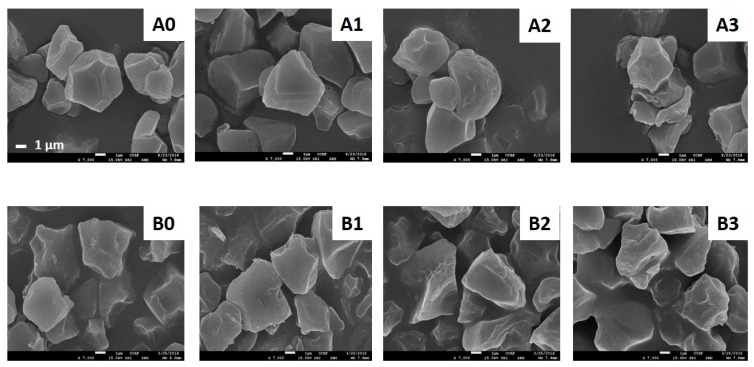
Scanning electron microscope images of OSA-modified Hopyeong (A) and Sinseonchal (B) rice starches (×7000). 1, 2, and 3 refer to the added percentages of octenyl succinic anhydride.

**Figure 3 molecules-24-00765-f003:**
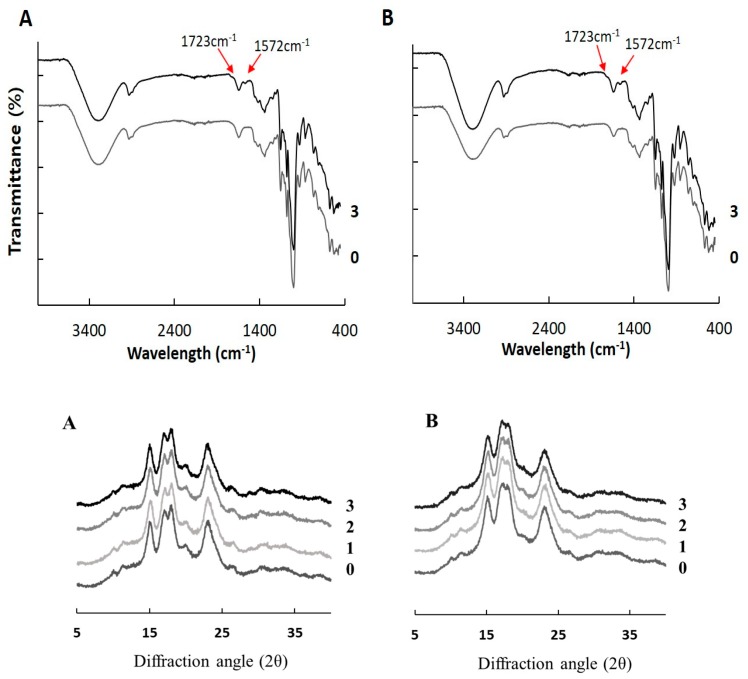
Fourier transform infrared spectroscopy and X-ray diffraction of OSA-modified. Hopyeong (A) and Sinseonchal (B) rice starches. 1, 2, and 3 refer to the added percentages of octenyl succinic anhydride.

**Figure 4 molecules-24-00765-f004:**
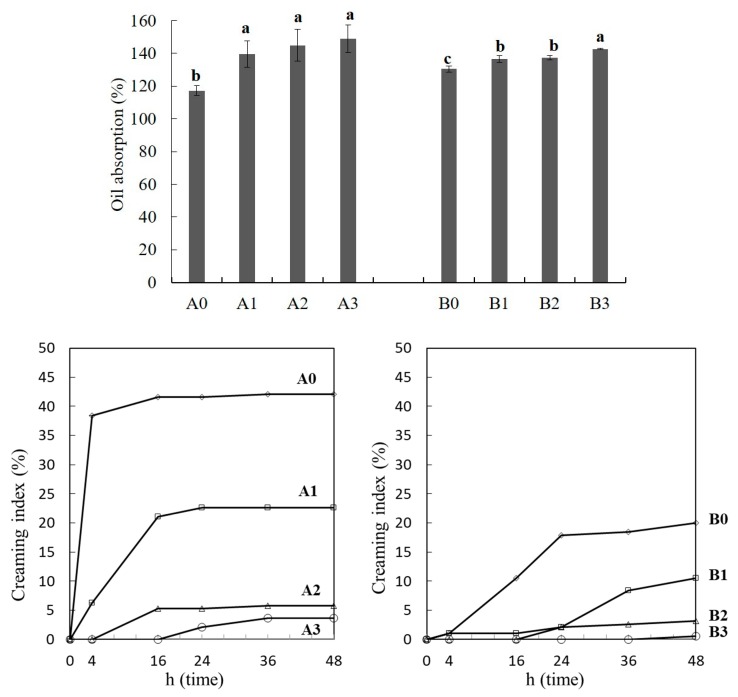
Oil absorption capacity and creaming index of OSA-modified Hopyeong (A) and Sinseonchal (B) rice starches. 1, 2, and 3 refer to the added percentages of octenyl succinic anhydride. Data represent as the mean ± SD; ^a–d^ Values accompanied in the same rice starch with different substitution ratios of OSA significantly differ (*p* < 0.05) by Duncan’s multiple range test.

**Table 1 molecules-24-00765-t001:** Amylopectin chain length distributions and in vitro digestibility of OSA modified rice starches.

OSA Substitution of Rice Starches	DS Value	Average Chain Length	Distribution (%)				In Vitro Digestibility			Relative Crystallinity (%)
			DP 6–12	DP 13–24	DP 25–36	DP ≥ 37	RDS (%)	SDS (%)	RS (%)	
Hopyeong 0%	-	21.78 ± 0.14 ^a^	27.20 ± 0.62 ^a^	45.08 ± 0.55 ^a^	12.33 ± 0.06 ^a^	15.40 ± 0.01 ^a^	61.85 ± 0.00 ^a^	34.96 ± 0.23 ^a^	3.20 ± 0.23 ^d^	36.89 ± 0.46 ^a^
Hopyeong 1%	0.0147 ± 0.0000 ^b^	21.20 ± 0.11 ^b^	27.77 ± 0.77 ^a^	45.78 ± 0.54 ^a^	12.26 ± 0.16 ^a^	14.18 ± 0.07 ^b^	60.22 ± 0.69 ^b^	35.00 ± 0.29 ^a^	4.78 ± 0.40 ^c^	36.26 ± 0.03 ^a^
Hopyeong 2%	0.0213 ± 0.0000 ^a^	20.82 ± 0.08 ^c^	28.41 ± 0.75 ^a^	46.20 ± 0.65 ^a^	12.05 ± 0.08 ^a^	13.34 ± 0.01 ^c^	58.64 ± 0.17 ^c^	32.15 ± 0.29 ^b^	9.21 ± 0.11 ^b^	35.41 ± 0.03 ^b^
Hopyeong 3%	0.0258 ± 0.0000 ^a^	20.79 ± 0.02 ^c^	28.99 ± 0.72 ^a^	45.75 ± 0.66 ^a^	11.94 ± 0.17 ^a^	13.31 ± 0.11 ^c^	49.86 ± 0.29 ^d^	35.08 ± 0.06 ^a^	15.06 ± 0.34 ^a^	34.42 ± 0.26 ^c^
Sinseonchal 0%	-	21.91 ± 0.32 ^a^	27.07 ± 0.76 ^b^	44.34 ± 0.22 ^a^	12.94 ± 0.09 ^a^*	15.65 ± 0.45 ^a^	67.46 ± 0.11 ^a^*	32.52 ± 0.34 ^a^*	0.03 ± 0.23 ^d^*	37.48 ± 0.02 ^a^
Sinseonchal 1%	0.0140 ± 0.0000 ^b^	20.76 ± 0.19 ^b^	29.94 ± 0.76 ^a^	44.42 ± 0.36 ^a^	12.07 ± 0.13 ^b^	13.57 ± 0.27 ^bc^	66.64 ± 0.11 ^a^*	32.23 ± 0.06 ^a^*	1.12 ± 0.06 ^c^*	33.08 ± 0.80 ^b^
Sinseonchal 2%	0.0246 ± 0.0000 ^a^	21.03 ± 0.05 ^b^	29.40 ± 0.17 ^a^	44.42 ± 0.29 ^a^	12.12 ± 0.05 ^b^	14.07 ± 0.06 ^b^*	62.86 ± 2.13 ^b^	35.08 ± 1.78 ^a^	2.06 ± 0.34 ^b^*	32.27 ± 0.15 ^b^*
Sinseonchal 3%	0.0262 ± 0.0000 ^a^	20.64 ± 0.10 ^b^	30.18 ± 0.51 ^a^	44.47 ± 0.28 ^a^	12.15 ± 0.13 ^b^	13.20 ± 0.10 ^c^	61.20 ± 1.61 ^c^	33.61 ± 1.32 ^a^	5.19 ± 0.29 ^a^*	30.78 ± 0.39 ^c^*

Data represent as the mean ± SD. ^a–d^ Values accompanied in the OSA rice starches with different substitution ratios of OSA significantly differ (*p* < 0.05) by Duncan’s multiple range test. * Significantly different between Hopyeong and Sinseonchal with same substitution ratios of OSA by t-test (* *p* < 0.05); 1%, 2%, and 3% refer to the substitution percentages of octenyl succinic anhydride modification of rice starches.

**Table 2 molecules-24-00765-t002:** Pasting and thermal characteristics of OSA rice starches using a rapid visco-analyzer and differential scanning calorimeter.

OSA Substitution of Rice Starches	Pasting Temp. (°C)	Viscosity (RVU)				DSC			
		Peak (P)	Final (F)	Breakdown (P-T)	Setback (F-T)	T_o_ (°C)	T_p_ (°)	T_c_ (°C)	∆H (J/g)
Hopyeong 0%	83.74 ± 0.62 ^a^	144.42 ± 0.47 ^d^	217.79 ± 7.48 ^d^	42.42 ± 1.30 ^d^	115.79 ± 8.31 ^b^	60.70 ± 0.19 ^a^	67.65 ± 0.12 ^a^	73.50 ± 0.18 ^a^	7.50 ± 076 ^a^
Hopyeong 1%	72.32 ± 0.05 ^b^	245.63 ± 1.83 ^c^	254.08 ± 1.41 ^c^	109.88 ± 2.06 ^c^	118.33 ± 1.18 ^b^	60.01 ± 0.40 ^a^	67.12 ± 0.14 ^b^	73.74 ± 0.18 ^a^	6.89 ± 0.36 ^a^
Hopyeong 2%	70.10 ± 0.59 ^c^	325.29 ± 1.00 ^b^	269.00 ± 0.12 ^b^	183.21 ± 0.77 ^b^	126.92 ± 0.12 ^ab^	59.09 ± 0.01 ^b^	66.73 ± 0.00 ^c^	73.86 ± 0.05 ^a^	6.65 ± 0.23 ^a^
Hopyeong 3%	68.75 ± 0.08 ^d^	393.29 ± 1.24 ^a^	280.42 ± 2.47 ^a^	247.33 ± 1.53 ^a^	134.46 ± 2.18 ^a^	58.06 ± 0.22 ^c^	66.12 ± 0.10 ^d^	73.69 ± 0.33 ^a^	7.35 ± 0.64 ^a^
Sinseonchal 0%	71.81 ± 0.21 ^a^*	279.92 ± 2.83 ^d^*	193.00 ± 0.59 ^d^	97.92 ± 3.06 ^d^*	11.00 ± 0.83 ^c^*	65.20 ± 0.06 ^a^*	70.96 ± 0.09 ^a^*	81.03 ± 3.17 ^a^	11.99 ± 3.16 ^a^
Sinseonchal 1%	71.00 ± 0.08 ^b^*	334.67 ± 2.59 ^c^*	208.92 ± 0.94 ^c^*	153.50 ± 3.77 ^b^*	27.75 ± 5.42 ^ab^*	61.74 ± 0.01 ^a^	67.78 ± 0.13 ^b^*	77.02 ± 0.67 ^a^	8.60 ± 0.40 ^a^ *
Sinseonchal 2%	70.98 ± 0.32 ^b^	352.46 ± 0.53 ^b^*	255.83 ± 0.71 ^b^*	125.75 ± 1.41 ^c^*	29.12 ± 1.59 ^a^*	59.42 ± 2.57 ^a^	66.27 ± 1.58 ^b^	76.30 ± 0.79 ^a^	7.66 ± 1.32 ^a^
Sinseonchal 3%	66.13 ± 0.21 ^c^*	492.96 ± 8.54 ^a^*	282.25 ± 4.01 ^a^	226.75 ± 5.66 ^a^	16.04 ± 6.89 ^bc^*	60.05 ± 1.52 ^a^	66.30 ± 1.28 ^b^	74.98 ± 1.16 ^a^	8.36 ± 0.02 ^a^

Data represent as the mean ± SD; ^a–d^ Values accompanied in the same rice starch with different substitution ratios of OSA significantly differ (*p* < 0.05) by Duncan’s multiple range test; * Significantly different between Hopyeong and Sinseonchal with same substitution ratios of OSA by t-test (* *p* < 0.05); 1%, 2%, and 3% refer to the substitution percentages of octenyl succinic anhydride modification of rice.

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
