# Peer review of "Properties and Digestibility of Octenyl Succinic Anhydride-Modified Japonica-Type Waxy and Non-Waxy Rice Starches"

_molecules, 2019, doi:10.3390/molecules24040765_

Reviewer 1 Report

In this paper, waxy and non-waxy rice starches were esterified with octenyl succinic anhydride (OSA) at different OSA levels, and the physicochemical, pasting, and thermal properties, the molecular structure of amylopectin, digestibility, and emulsion stability were investigated. However, there are many grammatical errors and badly structured sentences in the manuscript. The English language of the manuscript is needed to be polished. After major revision, it can be accepted for publication in Molecules.

Some comments:

1.      Lines 11-13: Please rewrite this sentence.

2.      Line 15: “gelatinzation” must be revised as “gelatinization”.

3.      Lines 15-16, please rewrite the sentence that “All OSA starches confirmed a new band at 1723 cm-1”.

4.      Lines 20-22: This sentence should be rewritten.

5.      Line 52, “OSA-modified japonica rice starches reported” should be revised as “OSA-modified japonica rice starches were reported”. And please check the font size of this sentence.

6.      Line 81, “Rice starch (50 g, dry basis, db) and distilled water (140 mL) added…” should be revised as “Rice starch (50 g, dry basis, db) and distilled water (140 mL) were added…”.

7.      Line 95, delete the redundant punctuation.

8.      Line 130, “Starch (2.5 g, 12% mb) mixed with distilled water” should be revised as “Starch (2.5 g, 12% mb) was mixed with distilled water”.

9.      Lines 129-134, please provide the supporting reference.

10.  Lines 135-139, please provide the supporting reference.

11.  Line 137, “Starch (3.0 mg, db) and deionized water (6.0 mg) put into…” should be revised as “Starch (3.0 mg, db) and deionized water (6.0 mg) were put into…”.

12.  Line 179, “he DS of OSA-modified starch” should be revised as “the DS of OSA-modified starch”.

13.  Line 227, “confirms” should be revised as “confirmed”.

14.  Line 240, delete the redundant punctuation.

15.  Lines 301-302, please check the expression of the sentence that “The increases in peak and breakdown viscosities due to OSA modification were higher in NRS than in WRS”. There are some grammatical errors.

16.  Lines 316-317, please check the expression of the sentence that “This result could explain that…”. There are some grammatical errors.

17.  Line 336, “OS-modified” should be revised as “OSA-modified”.

18.  Table 1 and Table 2, the data for all samples should be analyzed for significance of difference.

19. Figure 4, please provide a significance analysis of the data.

Author Response

Response to Reviewer 1 Comments

Point 1: Lines 11-13: Please rewrite this sentence

Response 1: Lines 10-12 rewrote that waxy and non-waxy rice starches from japonica type Korean varieties were esterified with different levels of octenyl succinic anhydride (OSA), and the molecular structure of amylopectin (AP), digestibility, and emulsion stability were investigated. (in red)

Point 2: Line 15: “gelatinzation” must be revised as “gelatinization”

Response 2: rewrote ‘gelatinization’ (in red)

Point 3: Lines 15-16, please rewrite the sentence that “All OSA starches confirmed a new band at 1723 cm-1”.

Response 3: Lines 14-15 rewrote that All OSA esterified starches showed a new band at 1723 cm-1’ (in red)

Point 4: Lines 20-22: This sentence should be rewritten

Response 1: Line 19-21 rewrote that it is suggested that amylose should have a higher affinity with OSA esterification than AP and that the emulsion stability should increase in WRS, but the digestibility should decrease in NRS after OSA modification. (in red)

Point 5: Line 52, “OSA-modified japonica rice starches reported” should be revised as “OSA-modified japonica rice starches were reported”. And please check the font size of this sentence.

Response 5: Line 52 rewrote that a previous study on OSA-modified japonica rice starches were reported, checked the font size (in red)

Point 6: Line 81, “Rice starch (50 g, dry basis, db) and distilled water (140 mL) added…” should be revised as “Rice starch (50 g, dry basis, db) and distilled water (140 mL) were added…”.

Response 6: Line 81 rewrote that rice starch (50 g, dry basis, db) and distilled water (140 mL) were added (in red)

Point 7:  Line 95, delete the redundant punctuation

Response 7: Line 95 deleted the punctuation (in red)

Point 8: Line 130, “Starch (2.5 g, 12% mb) mixed with distilled water” should be revised as “Starch (2.5 g, 12% mb) was mixed with distilled water”.

Response 8: Line 131 rewrote that starch (2.5 g, 12% mb) was mixed with distilled water (25 mL) (in red)

Point 9: Lines 129-134, please provide the supporting reference.

Response 9: Added reference (in red)

Point 10: Lines 135-139, please provide the supporting reference.

Response 10: Added reference (in red)

Point 11: Line 137, “Starch (3.0 mg, db) and deionized water (6.0 mg) put into…” should be revised as “Starch (3.0 mg, db) and deionized water (6.0 mg) were put into…”.

Response 11: Line 138 rewrote that starch (3.0 mg, db) and deionized water (6.0 mg) were put into (in red)

Point 12: Line 179, “he DS of OSA-modified starch” should be revised as “the DS of OSA-modified starch”.

Response 12: Line 180 rewrote that the DS of OSA-modified starch (in red)

Point 13: Line 227, “confirms” should be revised as “confirmed”.

Response 13: Line 227 rewrote that confirmed (in red)

Point 14: Line 240, delete the redundant punctuation.

Response 14: Line 237 rewrote the punctuation (in red)

Point 15: Lines 301-302, please check the expression of the sentence that “The increases in peak and breakdown viscosities due to OSA modification were higher in NRS than in WRS”. There are some grammatical errors.

Response 15: Line 302-304 rewrote that the WRS showed the higher peak viscosity than NRS peak viscosity. The increase of the OSA groups showed the increase of peak and breakdown viscosities. (in red)

Point 16: Lines 316-317, please check the expression of the sentence that “This result could explain that…”. There are some grammatical errors.

Response 16: Line 317-320 rewrote that this result could explain that the hydrophobic alkenyl groups weakened the internal hydrogen bonding of starch, enabling the starch to gelatinize at relatively low temperatures, and hence gradually decreasing the enthalpy of OSA starches with increased DS (in red)

Point 17: Line 336, “OS-modified” should be revised as “OSA-modified”.

Response 17: Line 334 rewrote that OSA-modified (in red)

Point 18: Table 1 and Table 2, the data for all samples should be analyzed for significance of difference.

Response 18: expressed significance analysis of the data (in red)

Point 19: Figure 4, please provide a significance analysis of the data.

Response 19: added significance analysis of the data in Figure 4. (in red)

Reviewer 2 Report

The manuscript reports the physico-chemical properties and digestibility of octenyl succinic 2 anhydride-modified japonica-type waxy and non-3 waxy rice starch. Data are interesting and properly treated statistically.

However, an extensive editing of English language and style is strongly required throughout the entire text. It has to be done by a native English speaker.

In this way, clarity would improve and the scientific merit would be even enhanced.

Author Response

Response to Reviewer 2 Comments

Point 1: The manuscript reports the physico-chemical properties and digestibility of octenyl succinic 2 anhydride-modified japonica-type waxy and non-3 waxy rice starch. Data are interesting and properly treated statistically. However, an extensive editing of English language and style is strongly required throughout the entire text. It has to be done by a native English speaker. In this way, clarity would improve and the scientific merit would be even enhanced.

Response 1: We edited sentence and English editing certificate is enclosed.

Reviewer 3 Report

The authors investigate the properties and digestibility of OSA-modified japonica-type waxy and non-waxy rice starches. The results can provide some information for application of OSA-modified starches in food systems. I have some comments as the below.

1. L73-74. It is OK to treat non-waxy starch with 0.2% NaOH, but the 0.2% NaOH can destroy the waxy starch according to my knowledge.

2. L158-161. It is better to give these data as Table or Figure, and add the SD and significant difference.

3. Figure 1. It is better to put A0-A3 and B0-B3 in one Figure. Please check the particle size data. According to the x-axis, the data is not so small.

4. Figure 3. FTIR. It is difficult to detect the peaks at 1723 and 1572 cm-1. It is better to present the regional spectrum, for example 2000 to 800 cm-1.

5. L249-251. The intensity is relative intensity. It is wrong to compare the peak intensity. The authors should compare their relative crystallinity.

6. Tables 1 and 2. The same “a” should be added to show the p>0.05.

Author Response

Response to Reviewer 3 Comments

Point 1: L73-74. It is OK to treat non-waxy starch with 0.2% NaOH, but the 0.2% NaOH can destroy the waxy starch according to my knowledge.

Response 1: In general, starch isolated by the alkaline steeping method with approximately 0.03-0.05 M NaOH solution (0.14-0.2%, w/v) achieved 73-85% yield of starch having 0.07-2.7% damaged starch (Puchongkavarin, Varavinit, and Bergthaller, 2005). Although this method can destroy rice starch granule, it seems to be the most effective method to separate starch form protein.

Point 2: L158-161. It is better to give these data as Table or Figure, and add the SD and significant difference.

Response 2: The data was represented in Table 1 (in blue)

Point 3: Figure 1. It is better to put A0-A3 and B0-B3 in one Figure. Please check the particle size data. According to the x-axis, the data is not so small.

Response 3: It is thought that all particles are mixed in various sizes because of agglomeration of the starch sample. However, the sauter mean diameter of OSA esterified starch is the same as shown.

Point 4: Figure 3. FTIR. It is difficult to detect the peaks at 1723 and 1572 cm-1. It is better to present the regional spectrum, for example 2000 to 800 cm-1.

Response 4: modified Figure 3

Point 5: L249-251. The intensity is relative intensity. It is wrong to compare the peak intensity. The authors should compare their relative crystallinity.

Response 5: added relative crystallinity data. 1. Added method in line 116-117, and L 249-251 rewrote represented relative crystallinity in Table 1 (in Blue)

Point 6: Tables 1 and 2. The same “a” should be added to show the p>0.05.

Response 6: expressed significance of difference (in red)

Round  2

Reviewer 1 Report

The authors have revised the manuscript carefully. It can be accepted for publication in Molecules according to the present form.